# Influence of Piezoelectric Properties on the Ultrasonic Dispersion of TiO_2_ Nanoparticles in Aqueous Suspension

**DOI:** 10.3390/mi12010052

**Published:** 2021-01-05

**Authors:** Seon Ae Hwangbo, Young Min Choi, Tae Geol Lee

**Affiliations:** 1Naosafety Team, Safety Measurement Institute, Korea Research Institute of Standards and Science, Daejeon 34113, Korea; hbsa@kriss.re.kr; 2Department of Thermal Systems, Korea Institute of Machinery and Materials, Daejeon 34103, Korea; anaud007@kimm.re.kr

**Keywords:** lead zirconate titanate, scanning electron microscopy, ultrasonics, nanoparticles

## Abstract

In this study, the soft-type and hard-type lead zirconate titanate (PZT) ceramics were compared in order to create an optimal system for ultrasonic dispersion of nanoparticles, and sound pressure energy for each PZT ceramic was analyzed and closely examined with ultrasonic energy. TiO_2_ was water-dispersed using the soft-type and hard-type PZT transducer, possessing different characteristics, and its suspension particle size and distribution, polydispersity index (PDI), zeta potential, and dispersion were evaluated for 180 days. Furthermore, it was confirmed that the particles dispersed using the hard-type PZT transducer were smaller than the particles dispersed using the soft-type PZT by 15 nm or more. Because the hard-type PZT transducer had a lower PDI, uniform particle size distribution was also confirmed. In addition, by measuring the zeta potential over time, it was found that the hard-type PZT transducer has higher dispersion safety. In addition, it was confirmed that the ultrasonically dispersed TiO_2_ suspension using a hard-type PZT transducer maintained constant particle size distribution for 180 days, whereas the suspension from the soft-type PZT aggregated 30 days later. Therefore, the hard-type PZT is more suitable for ultrasonic dispersion of nanoparticles.

## 1. Introduction

Well-distributed suspensions are useful in a wide variety of applications, such as food, medicine, paints, and fuel. Proper dispersion is crucial during the preparation, crushing, mixing, and formation of raw materials to prevent uneven distribution or the formation of particle agglomerates that are too large to stay in suspension. Therefore, the understanding and application of particle dispersion methodologies is important. The conventional physical and mechanical dispersion methods such as ball milling and jet milling contaminate the dispersion of nanoparticles due to high milling speed and ineffectiveness [1,2]. Interest in fundamental and applied research on nanoparticle dispersion using ultrasonic waves has been increasing. Cavitation used for ultrasonic dispersion is caused by sound pressure that repeats per second [3]. The inflow of sufficient sound pressure into the target liquid results in cavitation, which is caused by local pressure changes in the acoustics field and results in the formation, growth, vibration, collapse, and destruction of small bubbles or high-energy gas. The effect of cavitation yields sufficient energy to form bubbles in a liquid under negative pressure. Each bubble then develops in the rarefaction phase and collapses in the compression phase of the wave. Extreme temperature (more than 10,000 K) and pressure (more than 400 MPa) occur around the collapsing bubble [4,5,6,7,8]. The high energy generated during the collapse can be used to disperse particles in suspensions. However, aggregation control of nanoparticles in a colloidal state is insufficient when using the conventional bath or horn-type ultrasonic dispersion equipment for various reasons such as uneven dispersion performance due to unidirectional energy transfer and the inability to control heat generated when the equipment is operated. Therefore, it is necessary to develop ultrasonic dispersion technology for dispersing nanoparticles and an understanding of piezoelectric materials is essential.

Piezoelectric material develops a potential across its boundaries when subjected to a mechanical stress (or pressure) and vice versa. When an electric field is applied to the material, a mechanical deformation ensues. In particular, the converse piezoelectric effects could be used in ultrasonic dispersion equipment [9]. Based on the converse piezoelectric effect, an ultrasonic transducer produces cavitation effects. In particular, lead zirconate titanate (Pb[Zr_(1−x)_Ti_x_]O_3_ or PZT) is a suitable material for ultrasonic transducers because of its high piezoelectric constant, relative transients, and electromagnetic binding factors [10]. Such lead-based complex perovskites have been extensively investigated from both the academic and commercial perspectives. The most widely studied and used PZT configuration is located near the morphotropic phase boundary (MPB) between the tetragonal and rhombohedral phases. To meet the requirements of specific applications, the properties of PZT ceramics are usually improved using dopants. In general, donor additives induce “soft” piezoelectric behavior with high dielectric and piezoelectric activities suitable for sensor and actuator applications. In contrast, acceptor additives cause “hard” piezoelectric behavior that is suitable for ultrasonic motor applications [11,12,13,14,15,16,17,18,19,20]. It is important to select a piezoelectric material that is suitable for the specific field of application and to study the characteristics of the piezoelectric material depending on the specifications. The differences between the characteristics of soft and hard PZT imply that they exhibit different dispersion performances when used for ultrasonic dispersion. In this paper, ring-shaped piezoelectric ceramics made of hard PZT and soft PZT were used for ultrasonic dispersion.

Applications of ring-shaped transducers range from the latest scientific developments of scanning tunneling microscopy (STM) and atomic force microscopy (AFM) to medical diagnostics and therapeutics. Other applications include micro-positioners and vibration detection, ultrasonic pumps and liquid atomizers, phonograph pickups, and hydrophones [21]. A more suitable material for the ring-shaped ultrasonic system was selected via an ultrasonic dispersion experiment of TiO_2_ nanoparticles using the ring-shaped piezoelectric ceramics. After the dispersion of TiO_2_ nanoparticle colloid, the dispersion performance for each material was evaluated through particle size distribution and dispersion stability analysis for 180 days.

## 2. Experimental Procedure 

### 2.1. PZT Materials and Characterization

In this study, two commercially available PZT ceramics from DONGIL TECH. Co., Ltd. (Hwaseong, Korea) and Piezo Technologies (Indianapolis, IN, USA), respectively, were used for the ring-shaped transducers. Each transducer was manufactured from different PZT ceramics; detailed specifications are listed in Table 1. Scanning electron microscopy (SEM, Hitachi, Tokyo, Japan) micrographs of the two PZT ceramics were randomly acquired at 15 kV and magnifications of 3000× and 10,000×.

### 2.2. Polarization and Piezoelectric Properties

In this study, the ring-shaped PZT ceramics was customized after COMSOL simulation to determine the resonance mode and the electromechanical properties of the PZT ceramics that are used for ultrasonic transducers.

The circumference surfaces of the ring-shaped PZT ceramics were uniformly coated with silver paste to form electrodes and then annealed for 10 min at 600 °C, as shown in Figure 1. Each sample was then placed in a silicone oil bath at 120 °C for ceramic polarization, and an electric field of 4.5 kV mm^−1^ was applied for 30 min (poling time) to induce piezoelectric properties.

After sufficiently cooling the samples, the capacitance *C* (nF) of the ceramics was measured from the amount of electric charge *Q* (C) and the applied voltage *V* (V). The relative dielectric constant εr was calculated from the applied voltage, electrode area *A* (mm^2^) of the sample, and distance *L* (mm) of the two substrates according to the following Equation (1):*C* = *Q*/*V* = *ε*_r_*A*/*L*.(1)

To measure the electrical properties of the ring-shaped PZT ceramics, the electrodes were screen-printed with conductive silver paste [22]. Table 1 presented a list of the piezoelectric properties of the ceramics. The dielectric constant of hard PZT ceramics is significantly lower than that of soft PZT. Generally, PZT ceramics with a high dielectric constant degrade the polarizing efficiency and have low polarizing electric fields, leading to deterioration of the piezoelectric properties.

### 2.3. Experimental Setup

#### 2.3.1. Measurement of Ultrasonic Power

Ultrasonic treatment is typically performed in a liquid, with a metering method employed to calibrate its ultrasonic output power [23]. In this work, the amount of power used in ultrasound treatment was measured via calorimetry with distilled water as the medium, and the experimental setup is shown in Figure 2.

The PZT transducers were driven at a resonant frequency. The ring-shaped piezoelectric ceramic was fastened to aluminum housing for the dispersion test. In the process, the resonance point of each PZT changed. The resonance frequencies of the soft/hard PZT ceramics before and after fastening to the housing are shown in Table 2.

A 50 mL plastic container was filled with 4 mL of distilled water, and a thermocouple was inserted through a hole into the water in the container. The initial temperature of distilled water was 18 °C, and the temperature was recorded every 10 s until it stabilized. The experiment was terminated when the temperature of distilled water dropped below 60 °C. When heated through a PZT transducer for a certain period of time, the temperature of distilled water in the closed plastic container increased. The output, *Q*, of the system was calculated using Equation (2).
*Q* = *m*⋅*C*_p_⋅Δ*T*/Δ*t*,(2)
where *m* (g) is the total mass of water, *C*_p_ is the specific heat capacity of water (4.18 J/g·°C) and Δ*T*/Δ*t* (°C/s) represents the temperature gradient [24].

#### 2.3.2. Measurement of Acoustic Pressure

The acoustic pressure was measured using a hydrophone (Sonicheck-15, UL-TECH. Co., Ltd., Uiwang, Korea) near the distilled water at the center of the case, as shown in Figure 3. To determine the key point on which to optimize the ultrasonic dispersion system, a specific area with high ultrasonic activity was identified. Over the past decade, several researchers have measured the acoustic pressure to obtain the ultrasonic activation energy [25,26,27]. This information can be used to optimize future ultrasonic dispersion applications by selecting areas with high acoustic pressure and hence areas with high cavitation effects. In this work, the acoustic pressure was measured to study the acoustic energy of the ultrasonic dispersion system discussed.

#### 2.3.3. Ultrasonic Dispersion

The dispersion performance was evaluated by dispersing TiO_2_ nanoparticles by ultrasound treatment using a focused ultrasonic dispersion method in water without surfactants. In this method, acoustic energy is focused onto the center of a ring-shaped piezoelectric ceramic. Furthermore, as the driving temperature of the equipment is controlled using cooling water, nanoparticle dispersion up to the desired scale via long-time treatment is feasible. The amplified energy is transferred to the piezoelectric ceramic through the function generator and amplifier, and the sound energy is concentrated at the center of the ring. Particles are dispersed by passing the suspension through the focused ultrasonic wave, and because all the fluid circulates and passes through this section, a uniform particle dispersion is possible. In addition to controlling the heat that is generated during sonication, the cooling water circulating through the PZT transducer acts as a medium for generating ultrasonic waves and delivering sound pressure to the center of the colloid sample, thus allowing for the control of the dispersion conditions. Figure 4 shows a schematic diagram of the ultrasonic dispersion equipment for TiO_2_ nanoparticle, which is used in this study. The ultrasonic dispersion conditions are presented in Table 3, and the materials used for the ultrasonic dispersion experiment are detailed in Table 4.

After dispersion, the particle size distribution was analyzed using a centrifuge particle size meter (DC 24000, CPS Instruments, Prairieville, LA, USA). To measure the particle size, 1 wt.% of the sample was diluted 20 times with deionized water. Further, the zeta potential was measured using a size detector (Zetasizer Nano ZS, Malvern Panalytical, Malvern, UK).

## 3. Results and Discussion

### 3.1. Characterization of PZT Ceramics

Figure 5 shows SEM images of the two PZT ceramics. The soft PZT ceramic showed an inhomogeneous microstructure with grains measuring below 1 μm, while the hard PZT ceramic presented a more homogeneous microstructure with grains measuring up to 10 μm. Furthermore, the soft PZT ceramic exhibited slightly higher porosity than the hard PZT. Sokol et al. reported that the presence of fine grains reduces interparticle spacing, improving the piezoelectric properties [28].

### 3.2. Calculation of Ultrasonic Power by the Calorimetry Method

In general, the amount of energy used in ultrasound treatment can be determined by calculating the change in temperature of the target liquid caused by ultrasonic irradiation, which is typically evaluated via calorimetry. In this work, the delivered acoustic power was also determined calorimetrically [24,29,30,31,32], and the results of calorimetry using ultrasonic transducers with different resonant frequencies are shown in Figure 6. The quantity of heat was calculated using Equation (2). With the hard PZT transducer, the temperature increased rapidly in the initial 200 s and then stabilized. In contrast, the change in temperature with the soft PZT transducer was not significant. Thus, the hard PZT transducer is superior to the soft PZT in terms of ultrasonic power.

### 3.3. Measurement of Acoustic Pressure

Acoustic pressure measurements were performed to define the ultrasonic energy of the ultrasonic dispersion system. The measured voltage could be used to convert the output of the hydrophone into a pressure value using the expression.
*P* = *V*_measure_/*M*_sensitivity_,(3)
where *P* (Pa) is the acoustic pressure, *V*_measure_ is the voltage (mV) measured by the hydrophone and *M*_sensitivity_ is the sensitivity (mV/Pa) of the hydrophone. In fact, the sensitivity is the characteristic value of the materials, and thus the measured voltage could be proportional to the acoustic pressure. Figure 7 shows the measured voltage of the ultrasonic system at the resonant frequency of each PZT transducer. The tendencies for both PZT transducers were the same, i.e., the higher the applied voltage, the higher the measured voltage. However, under the same applied voltage (*V*_RMS_), the measured voltage values of the hard and soft PZT displayed a big difference. The value measured at the center of the ring-shaped piezoelectric ceramic for the hard PZT was about 20–110 mV, while the value measured for the soft PZT was less than 35 mV, showing relatively weak voltage compared to the hard PZT. This indicates that an increase in the input voltage (*V*_RMS_) does not result in a corresponding increase in the cavitation activity [33].

### 3.4. Focused Ultrasonic Dispersion of TiO_2_ Nanoparticles in Water

#### 3.4.1. Size Distribution

To confirm the dispersion performance of two PZT transducers, the particle size distribution of the TiO_2_ colloid after dispersion was determined at different centrifugal using a centrifuge particle size meter, which is more accurate than the conventional laser diffraction method [34] because in a colloid, the particle size distribution depends on both particle size and particle weight. Figure 8 shows the particle size distribution and the polydispersity index (PDI) before and after dispersion with both PZT transducers.

In Figure 8, the black line shows the particle size distribution before ultrasonic dispersion, revealing the amount of TiO_2_ nanoparticles agglomerated in water. This distribution was significantly wide, ranging from less than 100 nm to approximately 6 μm. After dispersion with PZT transducer, the average particle sizes were 64 and 51 nm, respectively. The PDI is a measure of the broadness of particle distribution; the larger the PDI, the broader the particle distribution. The PDI of the TiO_2_ solution before ultrasonic dispersion was 9.4. In contrast, the PDIs after ultrasonic dispersion were only 1.4 (hard PZT transducer) and 1.6 (soft PZT transducer), indicating that the particle size was more uniform owing to ultrasonic dispersion. The difference of 0.2 in these PDIs after dispersion indicates that the hard PZT transducer leads to a more uniform particle size than the soft PZT.

#### 3.4.2. Zeta Potential

To maintain the stability of the dispersed particles, they must be separated from electrostatic forces [35]. Thus, changes in colloidal stability were determined by measuring the zeta potential, which is a major indicator of colloidal dispersion stability [36,37]. It indicates the degree of electrostatic repulsion between adjacent similarly charged particles in a dispersed solution. A high zeta potential stabilizes small molecules and particles; therefore, a colloid with a high zeta potential, i.e., between ±40 and ±60 mV, can resist agglomeration, and the colloidal stability is considered to be excellent. If the zeta potential is low, i.e., less than ~35 mV [38], the attractive force exceeds the repulsive force, thereby breaking the suspension and leading to agglomeration. Figure 9 shows the zeta potential of the TiO_2_ colloid over time after dispersion using both PZT transducers.

The zeta potential in the case of ultrasonic dispersion using soft PZT transducer was approximately 43 mV, which was maintained for 30 days, before gradually decreasing to approximately 35 mV after 180 days. This implies that the repulsive force of the particles that prevents agglomeration was disturbed and that agglomeration could occur over time. However, when ultrasonic dispersion was performed using the hard PZT transducer, the zeta potential was approximately 48 mV, which slightly decreased over time and remained at 45 mV or more for 180 days. This was higher than the zeta potential of the sample dispersed using the soft PZT transducer. Based on these results, particle agglomeration in the sample dispersed using the hard PZT transducer can be concluded to have been less than that of the sample dispersed using soft PZT, and the dispersion stability of the hard PZT was superior.

#### 3.4.3. Stability of the TiO_2_ Colloids

The purpose of dispersion is to obtain a small, uniform hydrodynamic diameter of particles in a colloid [39], while increasing the duration of stability by preventing changes and re-agglomeration. Figure 10 tracks the changes in the particle size distribution over 180 days to evaluate the dispersion stability of the sample ultrasonically dispersed using both PZT transducers. Samples dispersed using the soft PZT transducer had a particle size distribution of approximately 65–68 nm immediately after dispersion, but some agglomeration was observed after 30 days, which coincided with the point when the value of the zeta potential became small. After 180 days, the average particle size was approximately 83–85 nm; this range is approximately 20 nm larger than the initial size, indicating that some particles agglomerated. Immediately after dispersion using the hard transducer, the average particle size was approximately 55 nm, and it remained at 50–55 nm for 180 days, indicating the particles did not agglomerate during this time. This result is similar to the zeta potential remaining at 45 mV or more for 180 days, in that, both sets of findings show the relatively high stability of samples dispersed using the hard PZT transducer.

## 4. Conclusions

In this study, the differences between the properties of two PZT transducers were evaluated, and each was used in an ultrasonic dispersion method to determine which is more suitable for particle dispersion in a nanoparticle suspension. Comparing their dispersion performance using a TiO_2_ nanoparticle suspension, the soft PZT transducer exhibited a mean particle size of approximately 60 nm after dispersion, whereas that of the hard PZT transducer was approximately 50 nm, which was approximately 10 nm smaller. In addition, the zeta potential was 10 mV higher for the hard PZT transducer than that of the soft PZT, indicating the former’s better dispersion stability. By observing the particle size change of TiO_2_ nanoparticle suspensions for 180 days, we confirmed that those dispersed by the hard PZT transducer did not agglomerate, indicating the excellent dispersion stability. Thus, the hard PZT transducer can be inferred to be more suitable for ultrasonic dispersion of nanoparticle suspensions than the soft PZT. The present study is expected to provide valuable insights into the development of efficient dispersion equipment and dispersion technologies.

## Figures and Tables

**Figure 1 micromachines-12-00052-f001:**
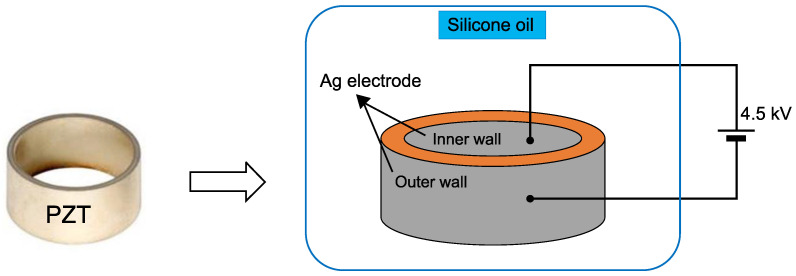
Poling process for ring-shaped PZT ceramic.

**Figure 2 micromachines-12-00052-f002:**
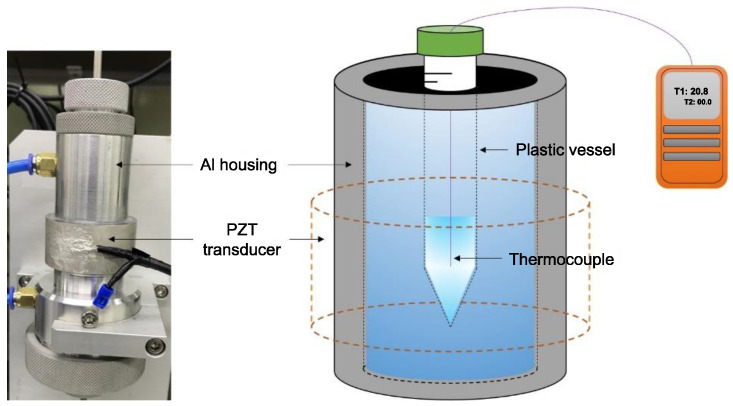
Photograph and schematic of the setup used to measure ultrasonic power by the calorimetry method.

**Figure 3 micromachines-12-00052-f003:**
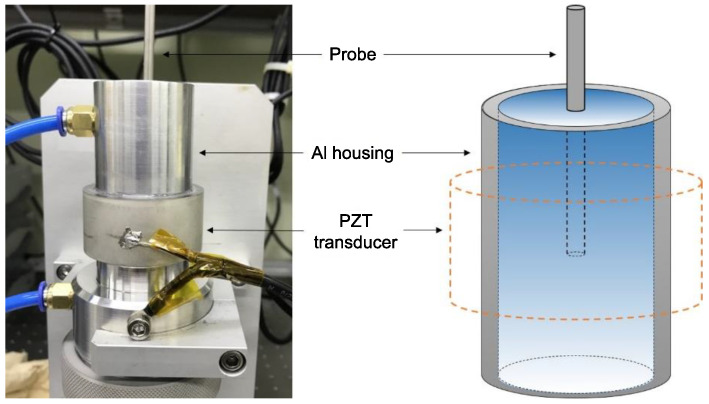
Photograph and schematic of the setup used to measure the acoustic pressure.

**Figure 4 micromachines-12-00052-f004:**
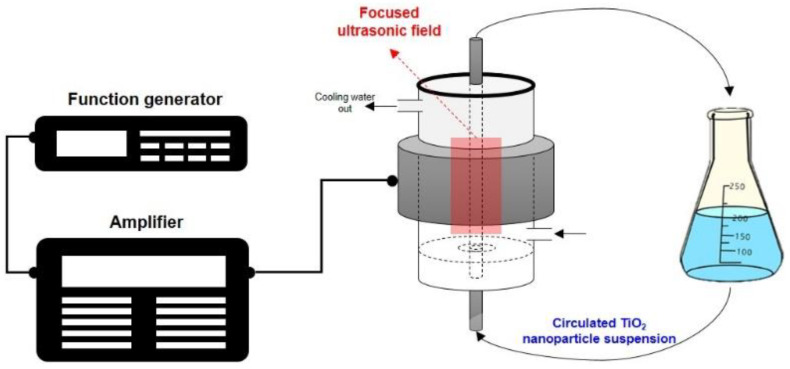
Schematic of the ultrasonic dispersion equipment for TiO_2_ nanoparticle in deionized water.

**Figure 5 micromachines-12-00052-f005:**
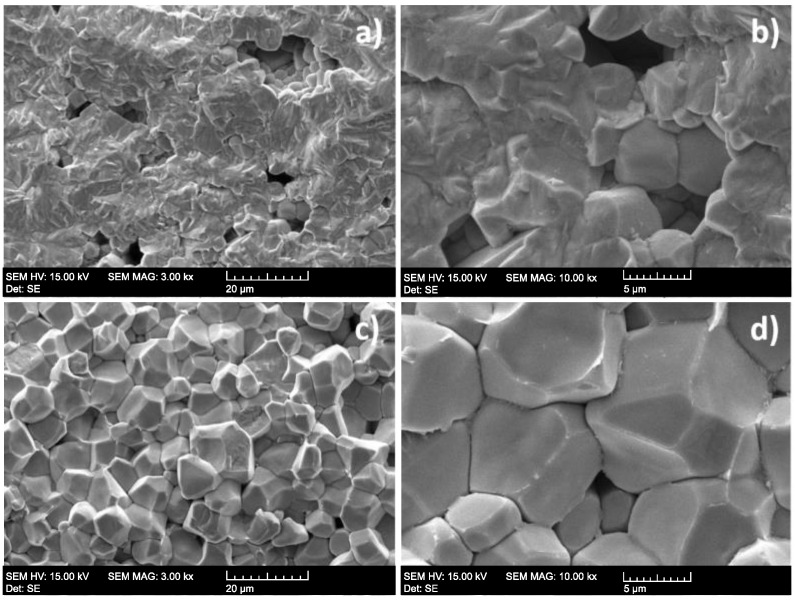
SEM images of fractured surfaces of (**a**) soft PZT ceramics (3000×), (**b**) soft PZT ceramics (10,000×), (**c**) hard PZT ceramics (3000×) and (**d**) hard PZT ceramics (10,000×).

**Figure 6 micromachines-12-00052-f006:**
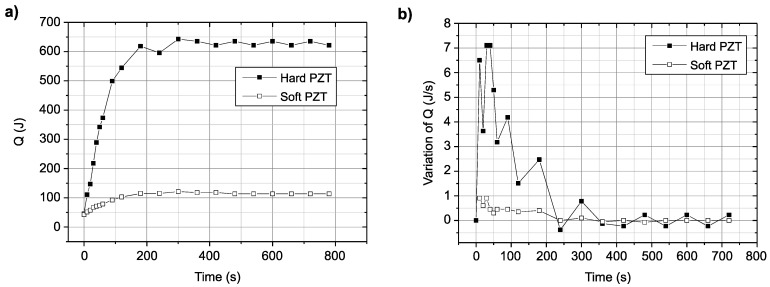
Variation in heat loss as a function of time for PZT transducer: (**a**) change in the total heat loss and (**b**) change in the heat loss per unit time.

**Figure 7 micromachines-12-00052-f007:**
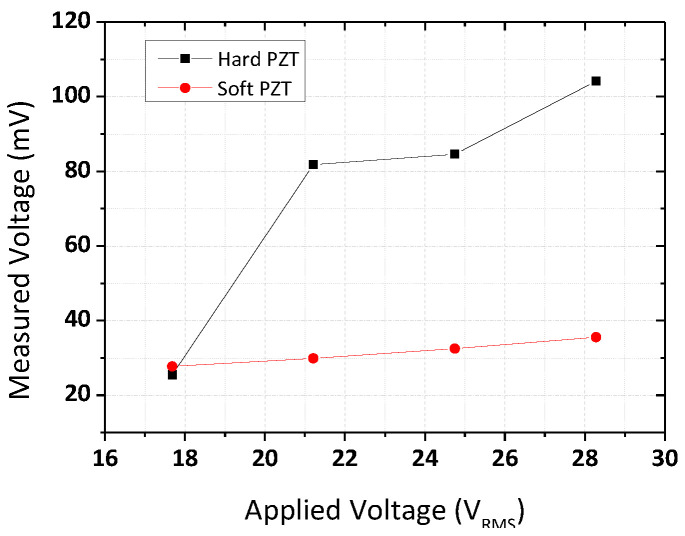
Variation in acoustic pressure delivered by the PZT transducer with voltage.

**Figure 8 micromachines-12-00052-f008:**
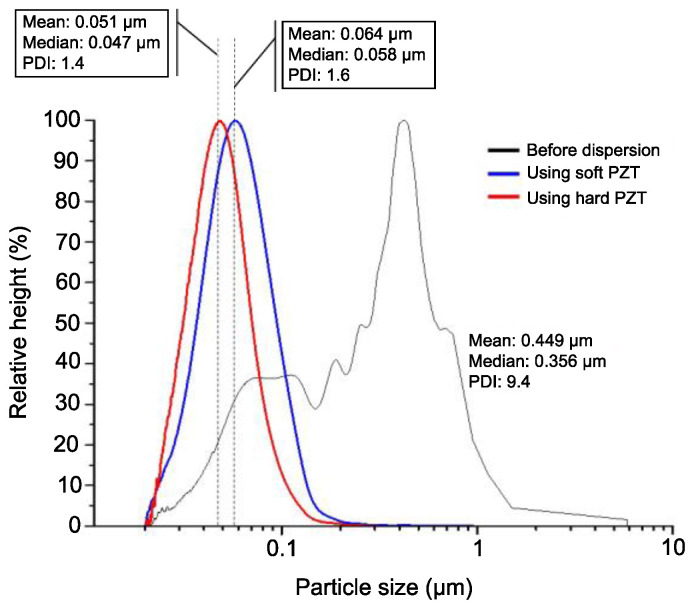
Particle size distributions and polydispersity indexes (PDIs) before and after dispersion using two PZT transducers (black: before dispersion, blue: after dispersion with soft PZT transducer, red: after dispersion with hard PZT transducer).

**Figure 9 micromachines-12-00052-f009:**
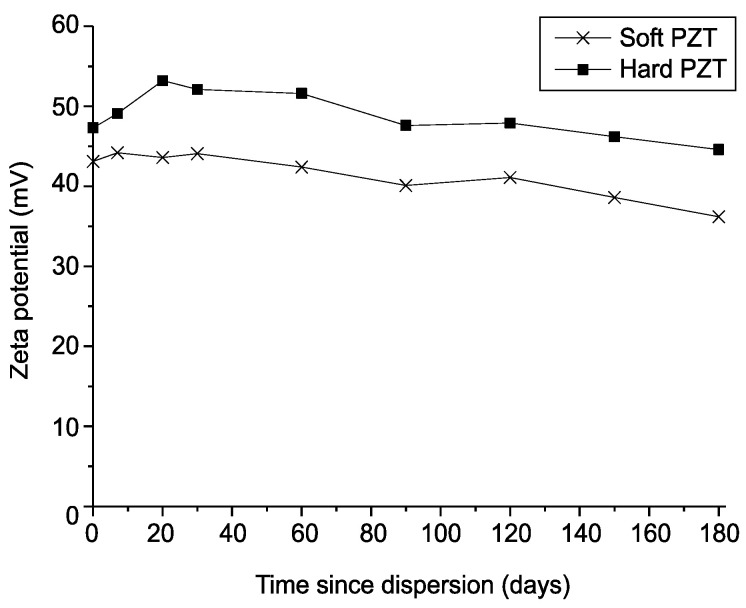
Zeta potential values of TiO_2_ colloids over time after dispersion using both PZT transducers (1 wt.%).

**Figure 10 micromachines-12-00052-f010:**
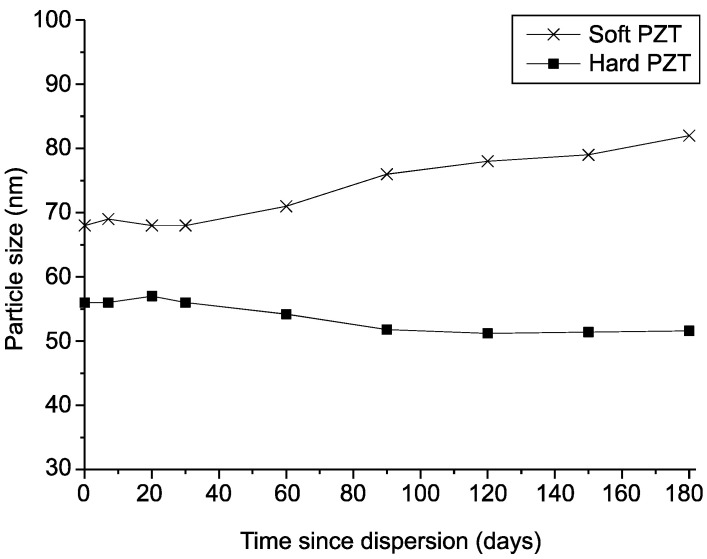
Variations in the particle size distributions of TiO_2_ colloids (1 wt.%) with time.

**Table 1 micromachines-12-00052-t001:** Specifications of ring-shaped lead zirconate titanate (PZT) transducers.

Characteristic	Hard PZT Ceramic (DONGIL TECH. Co., Ltd.)	Soft PZT Ceramic (Piezo Technologies)
Outer Diameter (mm)	49.90	49.96
Inner Diameter (mm)	39.90	39.82
Length (mm)	25.00	25.00
Frequency Constant (m/s)	2107.00	2035.61
Resonant Frequency (kHz)	421.40	401.50
Resonant Resistance (Ω)	1.30	6.09
Electro-Mechanical Coupling Factor (K_eff_)	0.310	0.262
Capacitance (nF)	8.90	11.72
Impedance (Ω)	1.81	16.08
Dielectric Constant (ε_33T_/ε_0_)	1439	1877
*tan δ	0.003	0.017

*tan δ is the loss tangent. For piezoelectric actuators, the loss tangent depends on the voltage amplitude.

**Table 2 micromachines-12-00052-t002:** Change of resonant frequency before and after housing connection.

Type	Resonant Frequency (kHz)	
	without Housing	with Housing
Hard PZT ceramic	421.40	396.2
Soft PZT ceramic	401.50	396.6

**Table 3 micromachines-12-00052-t003:** Experimental conditions used for ultrasonic dispersion.

Type	Applied Frequency (kHz)	Applied Power (W)	Exposure Time (min)
Hard PZT transducer	396.2	100	120
Soft PZT transducer	396.6	100	120

**Table 4 micromachines-12-00052-t004:** Physical properties of the nanoparticles, solvent, and suspension used in the experiment.

Nanoparticle	Liquid	Suspension
TiO_2_ (Degussa (Evonik) P25)	Deionized water	TiO_2_ colloid Concentration: 1 wt.% Volume: 100 mL Pre-treatment: none pH: 4.9–5.2
Mean diameter: 25 nm	Resistivity: 18.2 MΩ·cm	
Density: 3.78 g/cm^3^	pH: 7.2–7.6	

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
