# Peer review of "Influence of Piezoelectric Properties on the Ultrasonic Dispersion of TiO2 Nanoparticles in Aqueous Suspension"

_micromachines, 2021, doi:10.3390/mi12010052_

Round 1

Reviewer 1 Report

Hwoangbo and co-authors studied an ultrasonic method allowing the dispersion of nanoparticles (TIO2). The comparison between the use of soft and hard ceramics established the most suitable piezoelectric material for this kind of application. The paper offers significant information, but a few issues need to be addressed:

Line 33: “….methodologies is important….”

Are there any other method to obtain the dispersions? You should comment in the paper.

Line 34: “Ultrasonic vibrations are transmitted from nanoparticles”.

I’m confused with this statement. Are you saying that the nanoparticles generate ultrasound that causes physical effects such as cavitation?

Lines 43-44: “Piezoelectric materials convert electrical energy into mechanical strain and have been widely used in ultrasonic dispersion equipment”.

You should say that the piezoelectric materials can operate also exploiting the direct effect. Maybe, a reference would fit well. Here is a recent example:

  • doi: 10.3390/s20102800

Line 72: “Polarization and Piezoelectric Properties”

Why did you polarize the transducers? Did you buy them as unpolarized ceramics?

Line 98: “The PZT transducer tube was driven at the resonant frequency (approximately 400 kHz)”

Why did not use the exact resonant frequency for each transducer? Did you measure the acoustic power for both transducers? Did you use pulsed or continuous wave ultrasound for the measurement? My suggestion is to look at some recent works from the literature where the calorimetry has been applied for similar purposes. Here is an example:

  • doi:10.7863/ultra.16.02052

Lines 130-133: “In addition to controlling the heat that is generated during sonication, the cooling water circulating through the PZT transducer acts as a medium for generating ultrasonic waves and delivering sound pressure to the center of the colloid sample, thus allowing for the control of the dispersion conditions.”

I am confused with this sentence. Are you saying that ultrasound generated in the cooling water around the plastic container contributes to the dispersion of nanoparticles? How can it occur?

Line 158: “….Figure 5.”

In Fig. 5a, you show Q as a function of time. However, Q is not heat. Q [J/s] is the acoustic power estimated using calorimetry. You should revise those figures.

Line 158: “The quantity of heat was calculated using Equation (1).”

How did you calculate the heat from that equation?

Line 170: “…from 20 to 110 mV….”

Does the acoustic pressure has Volt as unit of measurement? You should also revise the related figure.

Author Response

Dear reviewer,

First of all, thank you for your valuable comments concerning our manuscript entitled "Influence of Piezoelectric Properties on the Ultrasonic Dispersion of TiO2 Nanoparticles in Aqueous Suspension" by Seon Ae Hwangbo, Young Min Choi and Tae Geol Lee for publication in Micromachines journal.

Your feedback and comments are all valuable and very helpful for revising and improving our paper. We have studied comments carefully and have made detailed and cautious English language corrections in addition to improving the quality of graphics used in manuscript and we hope that the revised manuscript meets your publication requirements.

Kind regards,

Dr. Seon Ae Hwangbo, Dr. Young Min Choi and Dr. Tae Geol Lee*

Reviewer 2 Report

The article is written consistently and clearly. It contains many descriptions of experiments and measurement results. However, it is necessary to highlight the novelty of the article. For higher paper quality, I suggest making the following improvements and clarifying inaccuracies.

  1. The literature cited in the article is outdated. More than half of the literature sources are older than 10 years. Especially the one used to define originality.
  2. The introduction does not reflect the novelty of the work.
  3. The choice of 400kHz resonant frequency is not clear (line 125), the authors cite their unpublished article, which cannot be verified.
  4. Ultrasonic dispersion or mixing is a very well-known phenomenon. Moreover comparison of hard and soft ceramics is redundant. It is well known that hard ceramics have high mechanical quality. Therefore, soft ceramics are used in sensors and hard ceramics are used in actuators. Therefore, novelty is not sufficiently expressed.
  5. Section 2.1 presents the commercial ceramics used for the experiments. However, section 2.2 already deals with production and polarization. Why polarization of the commercial ceramics is needed?

Author Response

(The authors gave the same response as above.)

Round 2

Reviewer 1 Report

The authors have made the necessary changes to the manuscript.

Author Response

Dear reviewer,

Thank you for your comments.